# Prediction of hyperbolic exciton-polaritons in monolayer black phosphorus

Fanjie Wang[1], Chong Wang[2,3], Andrey Chaves [4,5], Chaoyu Song[1], Guowei Zhang[6], Shenyang Huang[1], Yuchen Lei[1], Qiaoxia Xing[1], Lei Mu[1], Yuangang Xie[1] & Hugen Yan [1,7✉]

Hyperbolic polaritons exhibit large photonic density of states and can be collimated in certain propagation directions. The majority of hyperbolic polaritons are sustained in man-made metamaterials. However, natural-occurring hyperbolic materials also exist. Particularly, natural in-plane hyperbolic polaritons in layered materials have been demonstrated in $MoO_3$ and $WTe_2$, which are based on phonon and plasmon resonances respectively. Here, by determining the anisotropic optical conductivity (dielectric function) through optical spectroscopy, we predict that monolayer black phosphorus naturally hosts hyperbolic exciton-polaritons due to the pronounced in-plane anisotropy and strong exciton resonances. We simultaneously observe a strong and sharp ground state exciton peak and weaker excited states in high quality monolayer samples in the reflection spectrum, which enables us to determine the exciton binding energy of ~452 meV. Our work provides another appealing platform for the in-plane natural hyperbolic polaritons, which is based on excitons rather than phonons or plasmons.

[1] State Key Laboratory of Surface Physics, Key Laboratory of Micro and Nano-Photonic Structures (Ministry of Education), and Department of Physics, Fudan University, Shanghai 200433, China. [2] Centre for Quantum Physics, Key Laboratory of Advanced Optoelectronic Quantum Architecture and Measurement (MOE), School of Physics, Beijing Institute of Technology, Beijing 100081, China. [3] Beijing Key Lab of Nanophotonics & Ultrafine Optoelectronic Systems, School of Physics, Beijing Institute of Technology, Beijing 100081, China. [4] Departamento de Física, Universidade Federal do Ceará, Caixa Postal 6030, Campus do Pici, 60455-900, Fortaleza, Ceará, Brazil. [5] Department of Physics, University of Antwerp, Groenenborgerlaan 171, B2020, Antwerp, Belgium. [6] Institute of Flexible Electronics, Northwestern Polytechnical University, Xi'an 710072, Shaanxi, China. [7] Collaborative Innovation Center of Advanced Microstructures, Nanjing 210093, China. ✉email: hgyan@fudan.edu.cn

Excitons in atomically thin semiconductors are typically robust, showing large binding energy and high oscillator strength, due to the strong spatial confinement and reduced dielectric screening[1–5]. By introducing optical microcavities, excitons can couple with cavity photons, giving rise to microcavity exciton-polaritons (EPs). Such hybrid modes have been widely studied in monolayer transition-metal-dichalcogenides (TMDCs) films[6–8], providing a platform for exploring highly non-linear optics[9] and quantum fluids of polaritons[10] owing to their strong exciton-photon interactions. Besides cavity photons, excitons can also couple with free photons in the spectral range where the film has a positive imaginary part of the optical conductivity ($Im(\sigma) > 0$) due to the strong exciton resonances, leading to in-plane propagating EPs confined to the two-dimensional (2D) films. Such EPs were recently demonstrated in monolayer $WS_2$ encapsulated by hexagonal-boron-nitride[11], where the narrow exciton linewidth and large oscillator strength ensure $Im(\sigma) > 0$ needed for such in-plane polariton modes.

Similar to the phonon-polaritons[12] or plasmon-polaritons[13–15], the propagation characteristics of EPs can be potentially influenced by the anisotropy of the material. Especially, for a thin sheet with highly anisotropic excitons which exhibits sign-changing optical conductivity between the two in-plane crystallographic axes ($Im(\sigma_{xx}) \cdot Im(\sigma_{yy}) < 0$), the iso-frequency contours of the EPs can turn into hyperbolae. Compared with the closed iso-frequency contour, the hyperbolic dispersion theoretically supports an infinitely large wave vector accompanied by a change in the density of states from a finite to an infinite value[16–18]. Light-matter interaction is enhanced because of the hyperbolic iso-frequency contour, resulting in an enhancement of related quantum-optical phenomena, such as spontaneous emission[19,20]. While EPs with hyperbolic dispersion were reported in layer-structured hybrid perovskite crystals[21] for the out-of-plane propagating direction ($Re(\varepsilon) < 0$ in the in-plane direction, and $Re(\varepsilon) > 0$ in the out-of-plane direction), the in-plane hyperbolic exciton-polaritons (HEPs) confined to natural atomically thin thickness, which have higher binding energy and good tunability, still remain unexplored. Meanwhile, natural hyperbolic polaritons originated from other elementary excitations in 2D materials, such as phonons[12,22] and plasmons[23], have recently attracted tremendous attention. This also calls for the interrogation of hyperbolic polaritons based on excitons in 2D systems.

Recently, the in-plane anisotropic exciton absorption was observed in atomically thin black phosphorous (BP)[24,25]. The binding energy and oscillator strength increase with decreasing film thickness[26–28] and are expected to reach maxima at the monolayer limit. Moreover, the exciton resonance frequencies can be tuned by layer number[29,30], strain[31,32], and electrostatic gating[33–37], making atomically thin BP, especially the monolayer one, a promising platform for realizing tunable in-plane HEPs. However, limited by the challenges in sample preparation and preservation[38–40], there is even no agreement on the exciton binding energy for monolayer BP[25,29,41–44], let alone the exploration of the hyperbolicity. Thus, high-quality monolayer BP sheets are desirable to fulfill the promise. In this study, we determine the exciton binding energy of the high-quality monolayer BP by observing both the ground and excited states of the exciton and demonstrate the existence of in-plane propagating HEPs in it through a meticulous determination of the optical conductivity from the optical spectra. The photon energy window for HEPs shrinks with increasing sample thickness and disappears for samples thicker than 4-layer (L).

## Results

**Sample preparation and layer number confirmation.** High-quality monolayer and few-layer BP samples were mechanically exfoliated from single crystals, without any top capping layer or chemical treatment processes to maintain an ultra-clean surface. All-optical measurements were performed under effective nitrogen ($N_2$) purging at room temperature. Figure 1a shows the optical image of a BP flake exfoliated on a polydimethylsiloxane (PDMS) substrate. The monolayer area (marked by the blue dashed lines) was identified by the optical contrast (inset in Fig. 1a), and further confirmed by the reflection and photoluminescence (PL) spectra (Fig. 1b). Monolayer BP has an optical contrast of ~3%[44], much less than the ~10% contrast in monolayer TMDCs under the same condition (see Supplementary Fig. 3). As shown in Fig. 1b, prominent PL peak (black curve) and reflection peak (red curve) can be observed at around ~1.69 eV, consistent with the measurements by Li et al.[29]. The narrow linewidth (~50 meV) in the PL spectrum and the negligible Stokes shift (~6 meV redshift from the reflection peak), suggest of high-quality of our monolayer BP.

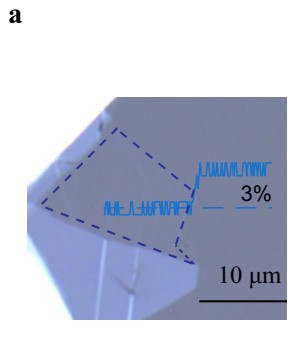
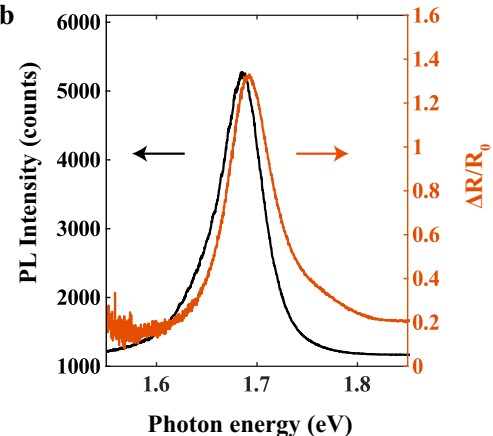

**Fig. 1 Optical characterization of monolayer BP. a** The optical image of BP on a PDMS substrate. The monolayer area is outlined by blue dashed lines. An optical transmittance line profile is included in the figure to highlight the optical contrast of ~3% for monolayer BP. **b** Reflection contrast spectrum (red curve) and photoluminescence (black curve) spectrum. The narrow linewidth (~50 meV) and the small Stokes shift (the energy difference between reflection and photoluminescence peak) of $\triangle_{Ref-PL} = 6$ meV, imply the high-quality of the monolayer BP sample.

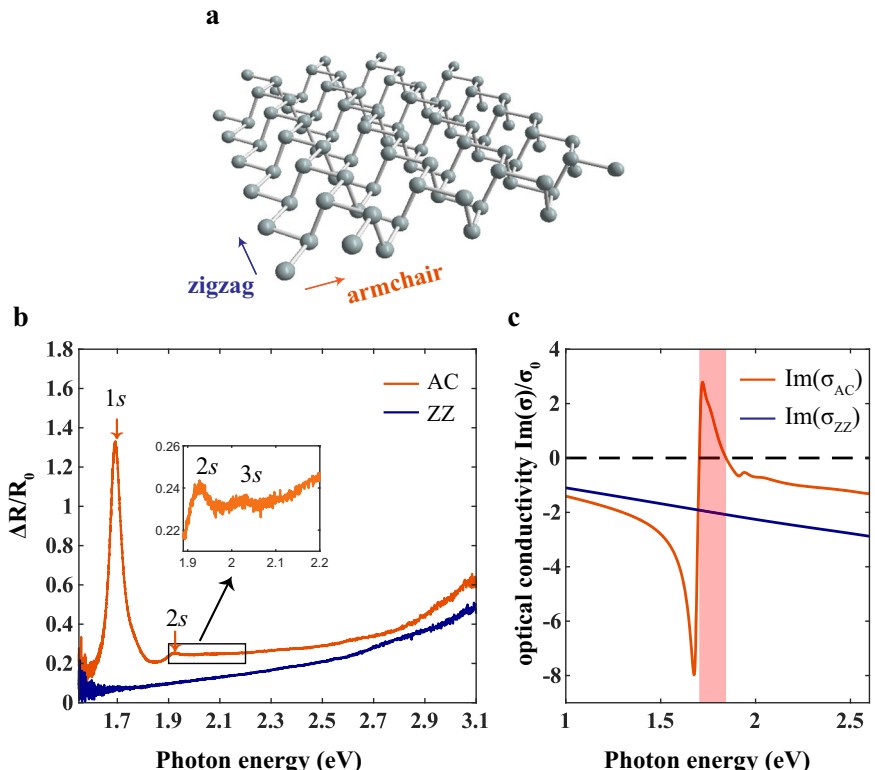

**Fig. 2 Exciton properties in monolayer BP. a** Schematic illustration of the crystal structure of monolayer BP. **b** Reflection contrast spectra of monolayer BP, with light polarized along with armchair (AC) and zigzag (ZZ) direction. The 1 s, 2 s and 3 s excitonic states are found at ~1.69, ~1.92, and ~2.04 eV, respectively. Inset: a zoom-in for the 2 s and 3 s resonance peaks. **c** The extracted imaginary part of the optical conductivity along with armchair and zigzag directions in the unit of $\sigma_0 = e^2/4\hbar$. The black dashed line at zero is a guide to the eyes. The highly anisotropic optical conductivity renders a HEP regime from 1.703–1.844 eV, as marked by the pink area, where $\text{Im}(\sigma_{AC}) \cdot \text{Im}(\sigma_{ZZ}) < 0$.

**Reflection spectra and the exciton binding energy.** The reflection contrast spectra of the monolayer BP under two different polarizations are shown in Fig. 2b, where $R$ and $R_0$ is the reflection from the sample and the bare substrate, respectively. The spectrum exhibits strong anisotropy and several exciton resonances peaks, though some of which are weak, can be observed when polarization is along the armchair direction. The strong lowest energy peak and the nearby weak peak are attributed to the 1 s and 2 s exciton resonances, in analogy to the hydrogenic Rydberg series. A 3 s exciton resonance with an even smaller oscillator strength is also observed, as shown in the inset. The energy differences are $\triangle_{12} = E_{2s} - E_{1s} = 230$ meV and $\triangle_{23} = E_{3s} - E_{2s} = 120$ meV, respectively. Note that the main features of the spectrum are reproducible on other high-quality samples (Supplementary Fig. 10). Based on a screened hydrogen model, where the nonlocal dielectric screening is taken into account, the binding energies in few-layer BP were calculated from the energy difference $\triangle_{12}$[26]. The same method can be applied to monolayer BP here, yielding an exciton binding energy of ~452 meV (See Supplementary Note II). In addition, a simpler estimation can be performed using the 2 s and 3 s resonance energies[2,3]. With increasing exciton quantum number $n$, the states are expected to be more Rydberg-like, which enables the estimation of the binding energy based on the traditional 2D hydrogenic model. In other words, we assume that 2 s, 3 s, and other higher exciton states follow the traditional 2D hydrogen model and only 1 s state deviates. Therefore, the binding energy of 2 s state is determined as $E_b^{2s} = (25/16)\triangle_{23}$ as a rough comparison, and the binding energy of 1 s state reads $E_b^{1s} = (25/16)\triangle_{23} + \triangle_{12} = 418$ meV, slightly smaller than the value obtained by the other method.

Previously, the exciton binding energy of monolayer BP was determined to be ~0.9 eV through photoluminescence excitation spectroscopy[25]. Other experiments indicate the binding energy is only ~0.1 eV if the monolayer BP is encapsulated by hexagonal-boron-nitride[29]. Our result here is only half of that in reference[25], though the dielectric constants of the substrates are similar. We argue that our determination of the exciton binding energy tends to be more reliable, based on the following reasons. First, the full width at half maximum (FWHM) of the PL peak we measured is ~50 meV, while it is ~150 meV (with the emission peak centered at ~1.3 eV mysteriously) in Wang's work[25,28], indicating our sample quality is better. Second, the measured peak energy (~1.69 eV) and obtained exciton binding energy (~452 meV) in our monolayer BP are fully consistent with the trend of the 2L-6L BP[26] studied by us previously, as manifested in Supplementary Fig. 5. Finally, the monolayer peak energy in our study is in good agreement with measurements by Li et al.[29].

**Hyperbolic exciton-polaritons.** Thanks to the high-quality of our monolayer BP, we observed sharp and strongly polarized exciton resonances. From the reflection spectra, the exciton binding energy was determined to be as large as ~452 meV. As a highly anisotropic 2D material with strong exciton resonance, monolayer BP is an ideal system for the realization of HEPs (See Supplementary Note IV for exciton absorption in BP).

The interaction of a 2D material with an external electromagnetic field can be captured by the sheet complex optical conductivity $\sigma$, which can be obtained from the reflection spectra[45,46] (see Supplementary Note I for details). $\sigma$ is usually

modeled by:

$$\sigma = \frac{i}{\pi}\frac{S_f}{\omega + i\tau_f^{-1}} + \frac{i}{\pi}\sum_{k=1}^{N}\frac{\omega S_k}{\omega^2 - \omega_k^2 + i\omega\gamma_k}, \qquad (1)$$

where $S_k$ is the spectral weight, $\omega_k$ is the transition frequency, and $\gamma_k$ is the resonance width of the $k_{th}$ inter-band transition resonance. $S_f$ and $\tau_f^{-1}$ are the Drude weight and scattering rate, respectively. In our case, however, the frequencies of inter-band transitions are much higher than the intra-band transitions (Drude response), thus only inter-band transitions (the second term in Eq. (1)) are considered. The inter-band transition is composed of a superposition of Lorentzian oscillators (the fitting details can be found in Supplementary Note I). Figure 2c shows the extracted imaginary parts of the optical conductivity along with the armchair and zigzag directions. Along the armchair direction, a spectral range with $\text{Im}(\sigma_{AC}) > 0$ can be observed between 1.703 and 1.844 eV, which defines the regime supporting in-plane EPs. In the meantime, $\text{Im}(\sigma_{ZZ})$ is negative throughout the whole spectrum range studied along the zigzag direction. Therefore, a region with a sign-changing $\text{Im}(\sigma)$ along the two directions exists ($\text{Im}(\sigma_{AC}) \cdot \text{Im}(\sigma_{ZZ}) < 0$), implying capabilities for monolayer BP to host in-plane HEPs, where the iso-frequency contours are hyperbolae. Such distinct dispersion behaviors are shown in Fig. 3 by calculating the EPs dispersion relation[23], using the loss function $-\text{Im}(1/\varepsilon)$, where $\varepsilon(\mathbf{q}, \omega)$ is the dynamical dielectric function. Specifically, $\varepsilon(\mathbf{q}, \omega)$ of a 2D system can be expressed as the following form:

$$\varepsilon(\mathbf{q}, \omega) = \varepsilon_{env} + \frac{i\sigma(\omega)}{\varepsilon_0\omega}\frac{q}{2} \qquad (2)$$

where the $\sigma(\omega)$ is the sheet optical conductivity, $\varepsilon_0$ is vacuum permittivity, and $\varepsilon_{env}$ is the dielectric constant of the environment in which the 2D sheet is embedded. The obtained loss function along the two principal axes is plotted as a pseudo-colour map in Fig. 3a, and the black dashed curve is based on the calculated loss function, which represents the energy at which the loss function peaks for each given wave vector. EPs with non-zero intensities can be found for $\mathbf{q}$ along the armchair direction, suggesting that EP resonance modes exist along the armchair direction in the hyperbolic regime, and the mode with $\mathbf{q}$ along the zigzag direction is not allowed. To quantify the behavior of exciton-

polaritons, the commonly used quality factor $Q = \frac{\omega}{\triangle\omega}$ can be calculated using the loss function results, where $\omega$ and $\triangle\omega$ are the resonance energy and linewidth[47] of the exciton-polaritons (details can be found in Supplementary Note III). At a near-zero wave vector, the Q value can reach as high as 35.52, which indicates a relatively low energy dissipation.

## Discussion

Now we can exam the characteristics of HEPs sustained by monolayer BP by taking a closer look at their dispersions. We assume that the wave vector $\mathbf{q}$ of HEPs is at an angle $\theta$ with respect to the armchair axis. The complex sheet optical conductivity at an angle $\theta$ in Eq. (2) thus takes the form $\sigma = \sigma_{AC}\cos^2\theta + \sigma_{ZZ}\sin^2\theta$. We rewrite Eq. (2) as:

$$\varepsilon(\mathbf{q}, \omega) = \varepsilon_{env} + \frac{i(\sigma_{AC}\cos^2\theta + \sigma_{ZZ}\sin^2\theta)}{\varepsilon_0\omega}\frac{q}{2}. \qquad (3)$$

At each frequency $\omega$, $\sigma_{AC}(\omega)$ and $\sigma_{ZZ}(\omega)$ values are obtained from the reflection spectra with light polarized along with the armchair and zigzag directions, respectively. The calculated loss functions based on Eq. (3) for fixed frequency $\omega$ and varying $\mathbf{q}$ are presented as pseudo-color maps in $\mathbf{q}$-space in Supplementary Fig. 7. Then, the maxima of the loss function in the map are extracted as the iso-frequency contour in $\mathbf{q}$-space, as shown in Fig. 3b. All of them exhibit hyperbolic shapes. The direction of the group velocity is the direction of the polariton energy flow. It usually doesn't align with $\mathbf{q}$ direction in anisotropic materials. Indeed, the direction of the group velocity is orthogonal to the $\mathbf{q}$-space iso-frequency contour[17,48]. The angles $\theta$ labeled in Fig. 3b are the directions of the asymptote of the iso-frequency curves. Therefore, the HEPs propagate at angles $\frac{\pi}{2} - \theta$ or $\frac{\pi}{2} + \theta$ with respect to the armchair direction. The appearance of an open iso-frequency curve will result in a singularity in the photonic density of states, where light-matter interaction is enhanced, and the wave vectors are much larger than those allowed in a vacuum. At each fixed energy, the HEPs with high momentum have nearly the same propagation directions (see arrows in Fig. 3b), so the energy flow in real space can be focused into narrow beams that do not spread laterally as they propagate. In the hyperbolic regime, with decreasing energy, the propagation direction of ray-like HEP approaches the armchair direction. Our simulations

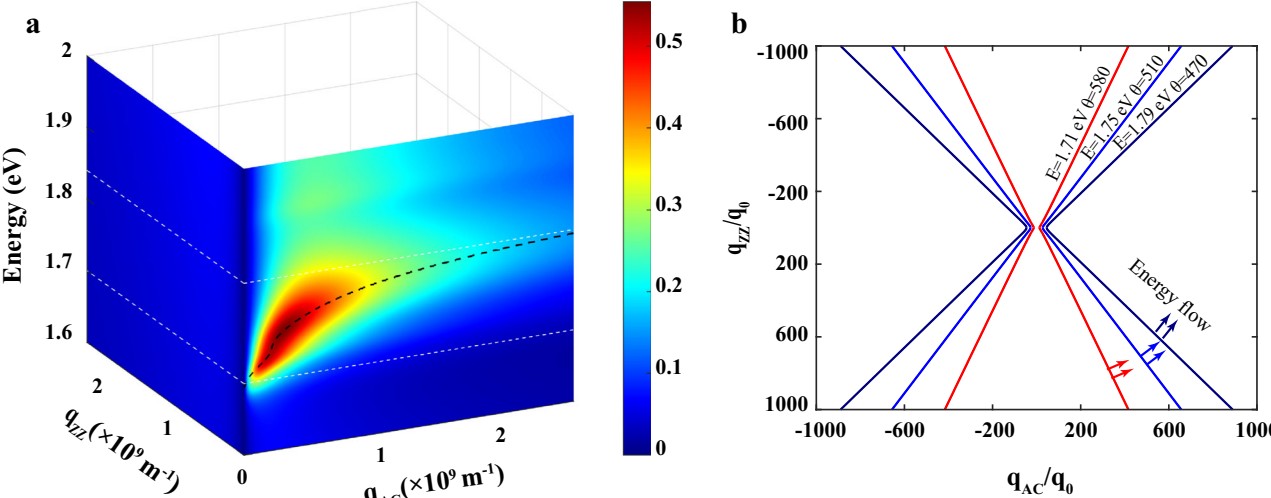

**Fig. 3 Hyperbolic exciton-polariton dispersion. a** The calculated loss function $-\text{Im}(1/\varepsilon)$ is displayed as a pseudo-color map. The EPs are highly anisotropic. The two white dashed lines mark the hyperbolic dispersion regime. The black dashed curve represents the dispersion. **b** Iso-frequency contours of EPs at different energies ($E = 1.71$ eV, 1.75 eV, 1.79 eV). The asymptote angle $\theta$ is labeled for each contour. The open iso-frequency contour indicates the hyperbolicity, with the propagating direction ($\frac{\pi}{2} - \theta$ or $\frac{\pi}{2} + \theta$) orthogonal to the iso-frequency contour, as indicated by the double arrows.

take into account realistic losses (the real part of optical conductivities) and the substrate effects. Indeed, for a higher loss material, the polariton itself is lossy, which reduces the field confinement[15]. In addition, we assume a frequency-dependent conductivity in the limit of $\mathbf{q} \rightarrow 0$. However, for an extremely confined polariton, the spatially-dispersive conductivity should be taken into account[13]. At a high wave vector, the polaritons decay faster in the free space. This will impose a wave vector cut-off to the open iso-frequency contour and close the otherwise open hyperbolic contour.

We emphasize that the HEP here is different from the polaritons predicted in references[14,15,49], where the occurrence of hyperbolic plasmon-polaritons in BP results from the interplay between intra-band and inter-band transitions. The HEPs discussed in this paper stem solely from the in-plane anisotropic excitonic behaviors, more specifically, from the sharp excitonic resonance with large oscillator strength and strong anisotropy, which creates a sizable energy window for the sign-changing optical conductivity. Note that the HEP is different from the hyperbolic exciton discussed in some literatures, which is referred to as the exciton associated with the saddle point of the electronic band[50]. Indeed, the oscillator strength of the exciton in monolayer BP is quite large. This can be manifested through comparisons. We did optical reflection measurements on several monolayer TMDCs, including $MoSe_2$, $WSe_2$, $WS_2$, and $MoS_2$. The extracted fitting parameters, including oscillator strength, peak energy and linewidth are summarized in Supplementary Table 1 and Table 2. As a reference, Supplemental Fig. 9 plots the energy differences $\triangle_{12}$ and the integrated optical conductivities (exciton absorption) as well.

Since the origin of the hyperbolicity is the strong anisotropic 1 s exciton resonance associated with the bandgap, one expects that the spectral range of hyperbolicity can be tuned with various means. Varying sample thickness is such an example. We obtained the optical conductivities of 2L-5L BP through their transmission spectra (Supplementary Fig. 2). Figure 4 presents the optical conductivity evolution from monolayer to four-layer BP, with the light polarized along the armchair direction. The hyperbolic regime, marked by the pink area, shifts from the visible to the near-infrared spectral range due to the decrease of the bandgap. Unavoidably, as layer number increases, the hyperbolic regime becomes narrower and disappears for samples thicker than 4 L, since the exciton oscillator

strength decreases[26,27]. Therefore, the in-plane hyperbolicity is only pertinent to atomically thin layers and the surface of a bulk BP lacks such properties. In addition, the $Im(\sigma_{AC}) > 0$ regime can be also tuned by strain and doping. The onset frequency of $Im(\sigma_{AC}) > 0$ should go up for tensile strain, because the peaks in the optical spectrum move to higher frequencies[31]. The onset frequency of $Im(\sigma_{AC}) > 0$ will increase for higher doping, as the first peak becomes less prominent due to Pauli blocking[33–36]. The exciton lifetime can also act as a crucial starting point to optimize the HEP properties. For example, a lower temperature[51] gives rise to EPs with a longer lifetime and increases light confinement. The instability of BP poses challenges in detecting the polaritons. Further experiments could be conducted on BP encapsulated by hexagonal-boron-nitride (hBN) through the Van der Waals assembly technique. Using scattering-type scanning near-field optical microscopy (s-SNOM), the real-space nanoimaging of hyperbolic exciton-polaritons can be performed and the polariton dispersion can be obtained by changing the incident light frequency. Alternatively, far-field absorption of periodically structured BP can also directly probe the exciton-polariton.

In summary, we obtain high-quality monolayer BP on a transparent PDMS substrate. From the experimental reflection contrast spectrum, we extract the optical resonance energies of excitonic 1 s, 2 s, and 3 s states, and thus determine the 1 s exciton binding energy. From the extracted complex optical conductivity, we are able to identify a $Im(\sigma) > 0$ regime along the armchair direction and show that monolayer BP is indeed a natural system to host potentially tunable in-plane propagating HEPs. Future efforts can be devoted to the near-field imaging and control of HEP beams in atomically thin BP systems[12].

## Methods

**Sample preparation.** BP flakes were obtained by mechanical exfoliation of bulk BP crystals with Scotch tape. A piece of commercially available bulk BP (HQ Graphene Inc.) was cleaved several times using Scotch tape. The tape containing the thin BP crystallites was then slightly pressed against a polydimethylsiloxane (PDMS) substrate (commercially available from Gelpak) and peeled off rapidly. Owing to the low optical contrast, the monolayer BP must be spotted under a microscope with a 50X objective. All the above steps were done inside a glovebox filled with $N_2$ ($O_2$ and $H_2O$ < 1ppm) to prevent degradation.

**Optical measurements.** The reflection and PL measurements of monolayer samples were carried out by using an Andor SR500i spectrometer in conjunction

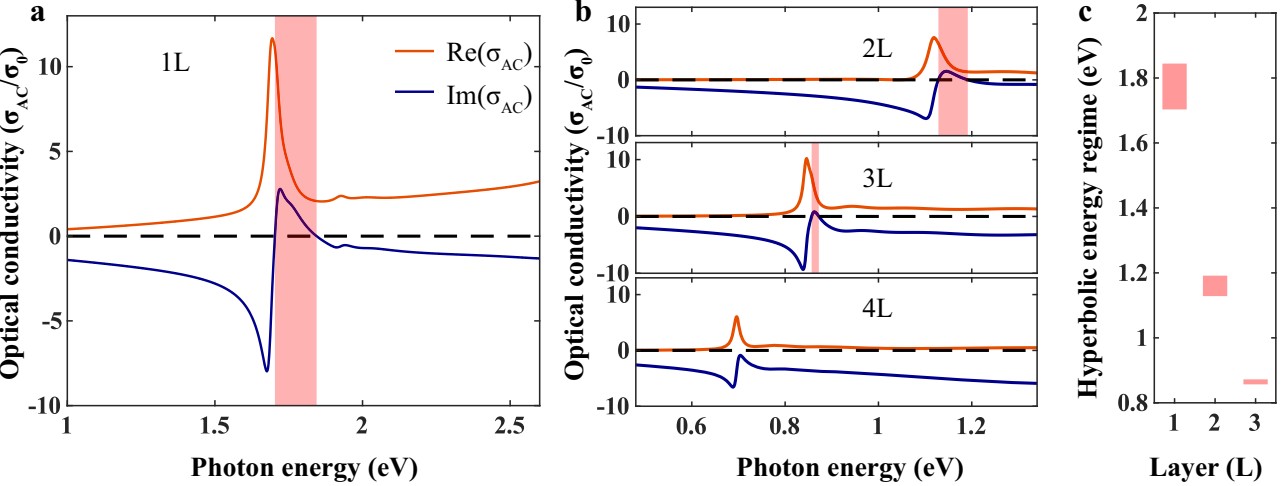

**Fig. 4 Thickness-dependent HEPs in atomically thin BP.** Extracted optical conductivity of atomically thin BP (**a**) 1L (**b**) 2L-4L. The hyperbolic regime is marked by the pink area. When the layer number increases, the area becomes narrower, and the onset shifts from the visible to the near-infrared range. **c** hyperbolic regime as a function of layer number.

with a Nikon Eclipse-Ti inverted microscope and a 50X objective. For reflection measurements, a tungsten halogen white light was used as the light source to cover the broad spectral ranges from 1.55–3.1 eV. The incident light was spatially filtered by a pinhole before focused onto the sample. The spot size on the sample is about ~5 μm, which is smaller than the sample size to make sure the light beam all goes through the sample. In this way, we exclude the possibility that the magnitude of the reflection contrast spectra is affected by the sample size. An analyzer was placed in front of the spectrometer entrance to allow for the analysis of the polarization dependence spectrum. The reflected light was finally collected by a cooled charge-coupled device camera. The measured reflected signals were analyzed in reflection contrast spectra, which is defined as $\triangle R/R_0 = (R − R_0)/R_0$, where $R$ and $R_0$ denote the light reflected from the sample and the bare PDMS substrate, respectively.

The polarization-resolved infrared measurements on few-layer BP were performed using a Bruker FTIR spectrometer (Vertex 70 v) equipped with a Hyperion 2000 microscope. A tungsten halogen lamp was used as the light source to cover the broad spectral range from 0.36–1.36 eV, in combination with a liquid nitrogen-cooled mercury-cadmium-telluride detector. The incident light was focused on BP flakes using a 15X IR objective, with the polarization controlled by a broadband ZnSe grid polarizer. The infrared extinction spectra were determined by $1 − T/T_0$ where $T$ and $T_0$ are the light transmittance from the sample and the bare PDMS substrate, respectively.

**Extraction of the optical conductivity**. The relation between the reflection contrast spectrum and the sheet optical conductivity ($\sigma$) is given by

$$\frac{R}{R_0} - 1 = \frac{\left|\frac{1 - n_s - Z_0\sigma}{1 + n_s + Z_0\sigma}\right|^2}{\left|\frac{1 - n_s}{1 + n_s}\right|^2} - 1. \tag{4}$$

The relation between the transmission contrast spectrum and the sheet optical conductivity is given by

$$1 - \frac{T}{T_0} = 1 - \left|\frac{1 + n_s}{1 + n_s + Z_0\sigma}\right|^2 \tag{5}$$

where $Z_0$ is the vacuum impedance, $n_s$ is the refractive index of the substrate ($n_s = 1.39$ for PDMS substrate). The optical conductivity contributed by excitons can be modeled by a superposition of Lorentzian oscillators. Supplementary Note I shows details on the extraction of the optical conductivity.

## Data availability

The data that support the findings of this study are available from the corresponding author upon request.

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

## Acknowledgements

H.Y. is grateful to the financial support from the National Natural Science Foundation of China (Grant Nos. 11874009, 11734007), the National Key Research and Development Program of China (Grant No. 2017YFA0303504), the Natural Science Foundation of Shanghai (Grant No. 20JC1414601), and the Strategic Priority Research Program of Chinese Academy of Sciences (XDB30000000). C.W. is grateful to the financial support from the National Natural Science Foundation of China (Grant No. 11704075). A.C. acknowledges financial support from the Brazilian Research Council (CNPQ), through the PQ and Universal programs, and from the Research Foundation-Flanders (FWO). G.Z. acknowledges the financial support from the National Natural Science Foundation of China (Grant No. 11804398), Natural Science Basic Research Program of Shaanxi (Grant No. 2020JQ-105), and the Joint Research Funds of Department of Science & Technology of Shaanxi Province and Northwestern Polytechnical University (Grant No. 2020GXLH-Z-026). S.H. acknowledges the financial support from the China Postdoctoral Science Foundation (Grant No. 2020TQ0078).

## Author contributions

H.Y. and F.W. initiated the project and conceived the experiments. F.W. prepared the samples, performed the measurements and data analysis with assistance from C.W., C.S. and S.H. A.C provided the theoretical support for the exciton binding energies. H.Y. and F. W. co-wrote the paper. H.Y. supervised the whole project. All authors commented on the paper.

## Competing interests

The authors declare no competing interests
