## [Peer Review File · Nature Communications]

Reviewers' Comments:

Reviewer #1:

Remarks to the Author:

The in-plane hyperbolic dispersion of polaritons in natural Van der Waals materials has enabled exciting possibilities in the field of anisotropic planar nano-optics, as well as stimulating fundamental studies. Doubtlessly, the topic is timely and very interesting.

The main claim of the authors in this work is the demonstration of in-plane hyperbolic exciton polaritons in atomically thin layers of black phosphorus. To this aim, the authors extract the in-plane optical conductivity along the armchair and zigzag crystalline directions from experimental reflectivity data and calculate a hyperbolic dispersion for in-plane exciton polaritons. Additionally, as expected (it is known that the bandgap of the material changes with the thickness of the crystal), a tunable thickness-dependent spectral range is shown.

In my opinion, although the reported results are interesting to some extent, and upon a careful reading of the manuscript, I believe that this manuscript lacks enough novelty to become publishable in Nature Communications. Please, find below my comments to support this conclusion along with assessments of the claimed novelties.

- It is widely known that black phosphorus is a material with in-plane highly anisotropic physical properties [Small Methods 1, 1700143 (2017); J. Phys. Chem. Lett. 6, 4280–4291 (2015); J. Am. Chem. Soc. 138, 300–305 (2016)]. Therefore, the claimed in-plane optical anisotropy is not surprising. Also, as mentioned by the authors, hyperbolic plasmon-polaritons have been demonstrated in black phosphorus.

- Although in-plane exciton polaritons are suggested in this work, the authors do not provide deeper insights into the polaritons' properties in black phosphorus: optical losses, lifetime values, visualization ...

- The claimed high excitons' binding energy is not surprising: it is widely known in atomically thin Van der Waals materials [Phys. Rev. Lett. 113, 076802 (2014), among many other works].

- As mentioned by the authors in the manuscript (page 4, line 63), the in-plane propagation of hyperbolic exciton polaritons in Van der Waals materials is yet to be explored. Unfortunately, this is not addressed in the current work.

Reviewer #2:

Remarks to the Author:

This paper reports on the reflection measurement of the monolayer BP films. BP is notoriously unstable, especially for the monolayer. However, the authors managed to obtain such high quality samples that the 2s and 3s exciton states can be observed, which are absent in other works as far as I know. This enables them to determine the exciton binding energy in monolayer BP, which was a controversy previously. I believe the result here for the exciton binding energy is the most convincing one. More importantly, by fitting the reflection spectra, the author can extract the optical conductivity and identify a sign-changing regime of the imaginary parts, which is the signature for the existence of in-plane hyperbolic exciton polaritons. For the natural hyperbolic materials, the natural phonon (MoO₃) and plasmon (WTe₂) in-plane hyperbolic polaritons have been reported. This paper provides the first experimental evidence for the existence of hyperbolic exciton polaritons in the natural 2D films. Unlike the MoO₃ and WTe₂, in which the hyperbolic regimes exist in the bulk as well, the in-plane hyperbolicity existing in the atomically thin BP layers cannot be found in bulk or relatively thick BP. The authors also demonstrate the changing of the hyperbolic regime in 2-4 layer BP films, indicating the strong tunability of the hyperbolicity in atomically thin BP films. Overall, the paper is scientifically sound and rigorous. The results are very interesting and consistent, and will open a field of natural hyperbolic exciton polaritons in 2D films. Given the huge interest on polaritons in 2D materials for the community, this paper is well-timed. I would recommend the paper for publication providing the authors properly comment on the following issues:

(1) The paper claims that when the layer number is 5 or larger, the hyperbolic regime disappears. It would be helpful if they can also show the optical conductivity data of a 5L and (or) 6L.

(2) Plasmons, phonons and excitons all deliver hyperbolic polaritons in certain materials. I recommend the authors discuss the difference between them, especially the features for hyperbolic exciton polaritons.

(3) I would recommend the authors add some guide lines for the zero points in Fig 4b. A summary of the hyperbolic regime vs layer number is recommended.

(4) I found that the q values in the hyperbolic regime can reach about 1000 times of q_0 , as shown in Fig 3. This is a quite large value. D Correas-Serrano (2016 J. Opt. 18 104006) discussed that a local conductivity may not be valid when q is large. I recommend the authors add some discussion about the possible non-local effect for the high q situation.

(5) In old literatures, hyperbolic excitons were coined to represent excitons associated to the hyperbolic electronic band structure, such as excitons around saddle points of the band. I believe here is not the case. To be more readable, this should be briefly mentioned in the paper.

Reviewer #3:

Remarks to the Author:

A very interesting manuscript on the prediction of hyperbolic polaritons in monolayer black phosphorus. In contrast to the previous works based on phonons and plasmons, this work utilizes excitons, adding an important member to the family of natural in-plane hyperbolic polaritons in layered materials. The mechanism of HEP is well explained and is supported by the reflectance contrast measurements performed on high-quality samples. This work paves new ways to observe HEP with semiconductor thin nanomaterials. I find the work suitable for Nature Communications. This version of the manuscript, on the other hand, needs to be improved by addressing the following issues.

Comments:

1 The author basically extracted the binding energies using 2D hydrogen model. In the previous paper: PRL 113, 076802 (2014), people have found the Rydberg series deviate from the hydrogen model, especially for the states with low quantum numbers. Could the author explain how reliable their reported value is? Actually, what is the error bar of the binding energy, and how does the author define the error? Considering they have to fit the reflection spectra and also the hydrogen model, the propagation error needs to be carefully extracted.

2 The authors have provided very solid calculations to demonstrate the existence of HEPs. I highly

recommend the authors add some comments on the future experiments: the possible ways to probe the dispersions of HEPs, considering the monolayer BP might be sensitive to nano-fabrications.

3 The x-axis of Figure 3a is sort of unfriendly to readers. Replacing it with a 3D hyperbolic dispersion will be much better.

4 In the legend of Fig 3. (Row # 190). "The black solid curve represents the fitted dispersion". The model for the fitting needs to be referred to.

5 All the quantitative analysis in the manuscript is from fitting the reflectance contrast spectrum. The author has to plot the fits over the raw data in Supplementary Figure 1 and list all the fitting parameters, which are very important to understand the impact of oscillator strength and linewidth.

Response to reviewers' comments

Reviewer #1 (Remarks to the Author):

The in-plane hyperbolic dispersion of polaritons in natural Van der Waals materials has enabled exciting possibilities in the field of anisotropic planar nano-optics, as well as stimulating fundamental studies. Doubtlessly, the topic is timely and very interesting.

The main claim of the authors in this work is the demonstration of in-plane hyperbolic exciton polaritons in atomically thin layers of black phosphorus. To this aim, the authors extract the in-plane optical conductivity along the armchair and zigzag crystalline directions from experimental reflectivity data and calculate a hyperbolic dispersion for in-plane exciton polaritons. Additionally, as expected (it is known that the bandgap of the material changes with the thickness of the crystal), a tunable thickness-dependent spectral range is shown.

In my opinion, although the reported results are interesting to some extent, and upon a careful reading of the manuscript, I believe that this manuscript lacks enough novelty to become publishable in Nature Communications. Please, find below my comments to support this conclusion along with assessments of the claimed novelties.

We sincerely appreciate the careful review from the reviewer. However, we disagree that our work lacks enough novelty to become publishable in Nature Communications. As reviewer 2 said: “Overall, the paper is scientifically sound and rigorous. The results are very interesting and consistent, and will open a field of natural hyperbolic exciton polaritons in 2D films. Given the huge interest on polaritons in 2D materials for the community, this paper is well-timed.” **And reviewer 3 said:** “A very interesting manuscript on the prediction of hyperbolic polaritons in monolayer black phosphorus. In contrast to the previous works based on phonons and plasmons, this work utilizes excitons, adding an important member to the family of natural in-plane hyperbolic polaritons in layered materials. This work paves new ways to observe HEP with semiconductor thin nanomaterials.” **Even the reviewer himself/herself noted** “The in-plane hyperbolic dispersion of polaritons in natural Van der Waals materials has enabled exciting possibilities in the field of anisotropic planar nano-optics, as well as stimulating fundamental studies. Doubtlessly, the topic is timely and very interesting.” **Hence, we believe our work is of great intrinsic interest to Nature Communication’s broad audience, which meets the acceptance criteria.**

Below, we would like to emphasize the novelty of our work in details.

(1) For the first time, we experimentally demonstrate 2D natural hyperbolic exciton-polaritons in high quality monolayer BP. 2D materials provide rich platforms for 2D polaritons. In particular, some hyperbolic polaritons have been found to exist naturally, such as phonon-polaritons in MoO₃ [Nature 562, 557-562 (2018)] and plasmon-polaritons in WTe₂ [Nat. Commun. 11, 1158 (2020)]. These materials exhibit strong in-plane anisotropy. Black phosphorus

(BP) shows anisotropic band structure, making it a potential candidate for hyperbolic material. The tunable hyperbolic plasmon-polaritons in BP have been theoretically predicted by Tony Low et al. [Phys. Rev. Lett. 116, 066804 (2016)] and Edo Van Veen et al. [Phys. Rev. A 12, 014011 (2019)]. However, up to date, there is no experimental verification for such plasmon-polaritons in BP. All aforementioned 2D polaritons are based on either phonons or plasmons. Here, for the first time, we experimentally demonstrate 2D natural hyperbolic exciton-polaritons in high quality monolayer BP, by fully determining the anisotropic dielectric function (optical conductivity) through optical spectroscopy.

(2) We directly observe the excited excitonic states, such as 2s and 3s states and determine the exciton binding energy. In the past, the exact binding energy of monolayer BP is controversial due to experimental challenges. In this work, we managed to prepare high quality monolayer BP sample, which enables us to observe multiple excitonic states, including ground state 1s and excited states 2s and even higher 3s state.

Shortly after the reintroduction of BP, Fengnian Xia's group in Yale determined the exciton binding energy in monolayer BP through photoluminescence excitation spectroscopy [Nat. Nanotechnol. 10, 517-521 (2015)]. The binding energy was claimed as high as ~ 0.9 eV for BP on SiO₂. However, this high value is inconsistent with later experiments by Feng Wang's group in Berkeley, which, together with calculations, estimated a much lower value [Nat. Nanotechnol. 12, 21-25 (2016)]. More crucially, Xia's experimental method to determine the binding energy is highly questionable. It relies on the observation of the onset of the continuum absorption, which is not supposed to be observable in such linear spectroscopy. Later on, our group determined the exciton binding energy for 2L-6L BP through the simultaneous observation of 1s and 2s excitonic states [Sci. Adv. 4, eaap9977 (2018)]. However, we didn't have monolayer results at that time.

Now, with the high quality of monolayer BP sample, using 2D non-hydrogenic model, we globally fit the data for 1L-6L BP, and the binding energy of monolayer BP was determined to be ~ 452 meV. We thus resolve a long-standing controversy regarding the exciton binding energy in monolayer BP. We believe this is the most reliable scheme to tackle the issue.

In summary, we believe that our results not only greatly enhance the understanding of the exciton properties in monolayer BP but also set up a platform for hyperbolic exciton-polaritons in anisotropic 2D systems.

- It is widely known that black phosphorus is a material with in-plane highly anisotropic physical properties [Small Methods 1, 1700143 (2017); J. Phys. Chem. Lett. 6, 4280-4291 (2015); J. Am. Chem. Soc. 138, 300-305 (2016)]. Therefore, the claimed in-plane optical anisotropy is not surprising. Also, as mentioned by the authors, hyperbolic plasmon-polaritons have been demonstrated in black phosphorus.

We agree with the referee that the in-plane anisotropic physical properties of black phosphorus have already attracted a lot of attention and many excellent studies have been performed. However, the anisotropy and hyperbolicity are conceptually very different. The fact is that the hyperbolic

exciton-polaritons in BP have not been either theoretically or experimentally investigated. Although the hyperbolic plasmon-polaritons in BP were theoretically predicted, the experimental evidence has not been reported yet. Our first study of hyperbolic exciton-polaritons in BP explores the extreme anisotropy in monolayer and few-layer BP, which is embodied by the sign-change of the optical conductivity along two axes ($\text{Im}(\sigma_{AC}) \cdot \text{Im}(\sigma_{ZZ}) < 0$). Such strict condition is not so easily achieved for many anisotropic materials. For instance, as demonstrated by our study, thick BP layers don't have hyperbolic exciton-polaritons, even though they still exhibit strong in-plane anisotropy. Therefore, nothing can be taken for granted and our first demonstration of hyperbolic exactions in BP represents a leap in the interrogation of anisotropic properties of BP.

As for the hyperbolic plasmon-polaritons raised by the referee, they are different type of polaritons compared to the exciton-polaritons, since they are originated from the collective oscillation of free carriers. Those studies regarding hyperbolic plasmon polaritons in BP don't compromise the novelty of ours. Most importantly, those studies are only theoretical predictions and no experimental demonstration has been realized.

- Although in-plane exciton polaritons are suggested in this work, the authors do not provide deeper insights into the polaritons' properties in black phosphorus: optical losses, lifetime values, visualization ...

We thank the referee for reminding us of this. In fact, we can gain insights on the polaritons' properties from our experimental data. In the revised version, we claimed that the sharp resonances absorption indicates a low loss. The largest quality factors $Q = \frac{\omega}{\Delta\omega}$ could reach up to 35. The reasonably large Q value indicates a low energy loss and the polaritons are long-lived. Using the loss function results (Fig. 3a in the main text), at a given wavevector, we plot the loss function versus frequency, and then the resonance peak and linewidth were extracted to calculate the quality factor Q, as shown in Supplementary figure 6.

In the revised main text, we added "To quantify the behavior of exciton-polaritons, the commonly used quality factor $Q = \frac{\omega}{\Delta\omega}$ can be calculated using the loss function results, where ω and $\Delta\omega$ are the resonance energy and linewidth⁴⁷ of the exciton-polaritons (details can be found in Supplementary Note □). At a near zero wave vector, the Q value can reach as high as 35.52, which indicates a relatively low energy dissipation."

Reference 47: Basov DN, Fogler MM, Garcia de Abajo FJ. Polaritons in van der Waals materials. Science 354, 195 (2016).

Supplementary Figure 6 | Quality factor of exciton-polaritons in monolayer BP. The exciton-polariton loss function at **a** $q = 1 \times 10^6 \text{ m}^{-1}$, **b** $q = 1 \times 10^8 \text{ m}^{-1}$, **c** $q = 1 \times 10^9 \text{ m}^{-1}$. The Q value is determined by $Q = \frac{\omega}{\Delta\omega}$. The Q value becomes smaller at larger wave vectors.

- The claimed high excitons' binding energy is not surprising: it is widely known in atomically thin Van der Waals materials [Phys. Rev. Lett. 113, 076802 (2014), among many other works].

We thank the referee for pointing out this. We agree with the referee that the high exciton binding energy is not so surprising. So, in the revised paper, we try not to emphasize that, even though it happens that the exciton binding energy of monolayer BP is almost the largest among all known 2D semiconductors. It does facilitate the relatively broader range of hyperbolic regime, as seen from the comparison to 2L and 3L BP. The major point of our paper is the existence of hyperbolic excitons in BP, which don't exist in TMDCs. Claim or not claim of the high exciton binding energy in BP are not so crucial to the main theme. We don't need to surprise readers with the high binding energy in the current study and we only objectively determined the value for monolayer through meticulous procedures.

- As mentioned by the authors in the manuscript (page 4, line 63), the in-plane propagation of hyperbolic exciton polaritons in Van der Waals materials is yet to be explored. Unfortunately, this is not addressed in the current work.

We agree on that the visualization of in-plane propagation of hyperbolic exciton-polariton needs further exploration. However, our work sets up a platform to observe hyperbolic exciton-polaritons in semiconductor thin films. The condition for the existence of in-plane propagation of the exciton-polariton is that the material should exhibit an optical conductivity whose imaginary part is positive. Thus, the fundamental question is whether the positive imaginary part of the optical conductivity could be achieved in the material. In this work, we experimentally determine the optical conductivity and predict such a spectral range.

Although we did not visualize the propagation of hyperbolic polaritons, we believe our results already manifest the basic properties of the hyperbolic polaritons in BP, such as the hyperbolic frequency range, the iso-frequency contour, the polariton dispersion relation, the quality factor and the layer-dependence. Visualization of the propagating polaritons can certainly give even more direct information on exciton polaritons. With suitable visible to near-infrared light coupled into

s-SNOM, we expect to see the polariton waves in real space. This can be done in the near future. However, we humbly believe it is out of the scope of current paper.

In the revised version, we add the following: “The instability of BP poses challenges in detecting the polaritons. Further experiments could be conducted on BP encapsulated by hexagonal boron nitride (hBN) through the Van der Waals assembly technique. Using scattering-type scanning near-field optical microscopy (s-SNOM), the real-space nanoimaging of hyperbolic exciton-polaritons can be performed and the polariton dispersion can be obtained by changing the incident light frequency. Alternatively, far-field absorption of periodically structured BP can also directly probe the exciton-polariton.”

Review #2

This paper reports on the reflection measurement of the monolayer BP films. BP is notoriously unstable, especially for the monolayer. However, the authors managed to obtain such high quality samples that the 2s and 3s exciton states can be observed, which are absent in other works as far as I know. This enables them to determine the exciton binding energy in monolayer BP, which was a controversy previously. I believe the result here for the exciton binding energy is the most convincing one. More importantly, by fitting the reflection spectra, the author can extract the optical conductivity and identify a sign-changing regime of the imaginary parts, which is the signature for the existence of in-plane hyperbolic exciton polaritons. For the natural hyperbolic materials, the natural phonon (MoO₃) and plasmon (WTe₂) in-plane hyperbolic polaritons have been reported. This paper provides the first experimental evidence for the existence of hyperbolic exciton polaritons in the natural 2D films. Unlike the MoO₃ and WTe₂, in which the hyperbolic regimes exist in the bulk as well, the in-plane hyperbolicity existing in the atomically thin BP layers cannot be found in bulk or relatively thick BP. The authors also demonstrate the changing of the hyperbolic regime in 2-4 layer BP films, indicating the strong tunability of the hyperbolicity in atomically thin BP films. Overall, the paper is scientifically sound and rigorous. The results are very interesting and consistent, and will open a field of natural hyperbolic exciton polaritons in 2D films. Given the huge interest on polaritons in 2D materials for the community, this paper is well-timed. I would recommend the paper for publication providing the authors properly comment on the following issues:

We sincerely appreciate the very positive comment and the endorsement from the reviewer.

- (1) The paper claims that when the layer number is 5 or larger, the hyperbolic regime disappears. It would be helpful if they can also show the optical conductivity data of a 5L and (or) 6L.

We thank for this useful advice. As mentioned by reviewer 3, the fitting process is extremely important. To get the best global fitting, we added more oscillators in our revised version, as has been done on TMDCs [Li Y, et al. Phys. Rev. B 90, 205422 (2014)]. The fitting results are shown in Supplemental Figs.1-3, which give an almost perfect fitting result. In this situation, the monolayer BP yields a $\epsilon_{AC}(0) = 12$ along the armchair direction. Considering that the optical

conductivities of BP is anisotropy, the $\epsilon_{zz}(0) = 10$ was applied along the zigzag direction. It is consistent with the reported values [A. Castellanos-Gomez et al., 2D Materials 1, 025001 (2014)], where $\epsilon_{AC} = 12.5$ and $\epsilon_{ZZ} = 10$. We find that when the layer number is 4L, the hyperbolic regime disappears. In the past version ($\epsilon_{AC}(0) = 8.3$), the hyperbolic regime for 4L exists, although with a very tiny range. By unifying the dielectric constant at zero frequency for all layers ($\epsilon_{AC}(0) = 12$), the small spectral regime of 4L BP disappears now. Of course, we examined the 5L BP, together with other thicker samples, the hyperbolic regime does not exist either, as shown in Supplementary Fig. 2.

(2) Plasmons, phonons and excitons all deliver hyperbolic polaritons in certain materials. I recommend the authors discuss the difference between them, especially the features for hyperbolic exciton polaritons.

We thank the reviewer for pointing out this. The theoretically reported hyperbolic plasmon-polaritons (anisotropic collective electronic excitation) in BP result from the interplay between intra-band and inter-band transitions, it covers a relatively narrow spectral range from the mid- to the far-infrared.

The hyperbolic phonon-polaritons are the coupling of photons with collective optical phonon. They usually exhibit low loss. However, the hyperbolic frequency range is hard to tune.

The hyperbolic exciton-polaritons stem from the in-plane anisotropic excitonic excitation, and it has higher and wider frequency resonances.

The tightly bound excitons in layer 2D material leads to particularly strong coupling of excitons to photons. The exciton coupling to the light field is the essential in the concept of the exciton-polaritons. Once the exciton resonance is anisotropic, the realization of opposite imaginary part of optical conductivities between two crystalline directions is possible, which induces a hyperbolic exciton-polariton mode. More interesting, the excitons are more amenable to external controls. For example, 1% in-plane strain changes the interlayer coupling of few-layer BP almost 10%. Other methods like doping and changing thickness are also efficient to tune the exciton response. The external effects on the excitons can be directly transmitted to hyperbolic exciton-polaritons, consequently, the hyperbolic exciton-polaritons share the same tunable properties as excitons.

(3) I would recommend the authors add some guide lines for the zero points in Fig 4b. A summary of the hyperbolic regime vs layer number is recommended.

We are extremely grateful for this advice. In the revised version, we added guide lines for the zero points in Fig. 4b, and added a Fig. 4c to summarize the hyperbolic range vs layer number.

(4) I found that the q values in the hyperbolic regime can reach about 1000 times of q_0 , as shown in Fig 3. This is a quite large value. D Correas-Serrano (2016 J. Opt. 18 104006) discussed that a

local conductivity may not be valid when q is large. I recommend the authors add some discussion about the possible non-local effect for the high q situation.

We thank the reviewer for the recommendation. We indeed ignored the non-local effect in our paper. Throughout our paper, we assumed a frequency-dependent conductivity in the limit of $q \rightarrow 0$, which we believe is sufficient for most experimental and theoretical scenarios.

For the extremely confined polaritons in atomically thin BP films, the local conductivity may be not complete. When taking into account spatially-dispersive conductivity, the hyperbolic topology dispersion may be destroyed for a significant range of wave vectors. More precisely, at lower q_{AC} , the polariton still keep a true open hyperbolic topology, but as q_{AC} increases, the influence of nonlocality becomes more apparent. The topology changes drastically into a closed topology when q_{AC} get close to a certain value, it occurs because high- q waves decay faster in the free-space.

The related sentence “In addition, we assume a frequency-dependent conductivity in the limit of $q \rightarrow 0$. However, for an extremely confined polariton, the spatially-dispersive conductivity should be taken into account¹³. At a high wave vector, the polaritons decay faster in the free-space. This will impose a wave vector cut-off to the open iso-frequency contour and close the otherwise open hyperbolic contour.” has been added in the manuscript.

Reference 13: Correas-Serrano D, Gomez-Diaz JS, Melcon AA, Alù A. Black phosphorus plasmonics: anisotropic elliptical propagation and nonlocality-induced canalization. *J. Opt.* 18, 104006 (2016).

(5) In old literatures, hyperbolic excitons were coined to represent excitons associated to the hyperbolic electronic band structure, such as excitons around saddle points of the band. I believe here is not the case. To be more readable, this should be briefly mentioned in the paper.

We thank the reviewer for pointing out this. Hyperbolic excitons occur at saddle point of the electronic band, where the constant energy surface near critical points are hyperboloids. In other words, it's a description of the electronic band structure and has nothing to do with the polariton dispersion.

To clarify it, we added the sentence “Note that the HEP is different from the hyperbolic exciton discussed in some literatures, which is referred to the exciton associated with the saddle point of the electronic band⁵⁰” in our manuscript.

Reference 50: Yu Y, Cardona M. *Fundamentals of Semiconductors: Physics and Materials Properties*. (Springer, New York, 2010).

Reviewer #3 (Remarks to the Author):

A very interesting manuscript on the prediction of hyperbolic polaritons in monolayer black phosphorus. In contrast to the previous works based on phonons and plasmons, this work utilizes excitons, adding an important member to the family of natural in-plane hyperbolic polaritons in layered materials. The mechanism of HEP is well explained and is supported by the reflectance contrast measurements performed on high-quality samples. This work paves new ways to observe HEP with semiconductor thin nanomaterials. I find the work suitable for Nature Communications. This version of the manuscript, on the other hand, needs to be improved by addressing the following issues.

We sincerely appreciate the very positive comment and the endorsement from the reviewer.

Comments:

1 The author basically extracted the binding energies using 2D hydrogen model. In the previous paper: PRL 113, 076802 (2014), people have found the Rydberg series deviate from the hydrogen model, especially for the states with low quantum numbers. Could the author explain how reliable their reported value is? Actually, what is the error bar of the binding energy, and how does the author define the error? Considering they have to fit the reflection spectra and also the hydrogen model, the propagation error needs to be carefully extracted.

We thank the referee for this important advice. Firstly, we need to point out that the binding energy of monolayer BP (~ 452 meV) is actually obtained using the 2D non-hydrogenic model mentioned by the referee, as explained in the Supplementary Note □. We use the energy difference between the peaks assigned to $1s$ and $2s$ excitonic states to calibrate the model, which gives us information on the effective dielectric screening of the BP layer in the Rytova-Keldysh potential that we consider for the electron-hole interaction. We then numerically solve Schrodinger equation within the Wannier-Mott model, assuming the Rytova-Keldysh interaction potential, to obtain the (non-hydrogenic) exciton Rydberg series. In the main text, we assumed that $2s$ and $3s$ (but not the $1s$) states approximately follow the hydrogenic form just to obtain a rough estimation of the binding energies and to compare it to our numerically obtained results, but the actual binding energy (~ 452 meV) we claimed for the ground state exciton in monolayer BP is derived from the non-hydrogenic model, which we believe is more reliable, as suggested by the referee. This is clarified in the current version of our paper, and we apologize for this misunderstanding.

The reliability of our results for binding energies is supported by several arguments: (i) the Wannier-Mott model we used, along with the Rytova-Keldysh electron-hole interaction potential, has been successfully applied for the study of excitons in 2D materials in several previous papers (including the paper mentioned by the referee); (ii) the numerical solution we used is based on the diagonalization of the exciton Hamiltonian, discretized in a finite difference scheme, assuming a non-uniform (logarithmic) mesh which accounts both for the fine details close to the exciton origin, as well as for the decay of the exciton wave function far from the origin, which thus guarantees the numerical accuracy of the obtained energies; (iii) the binding energies obtained here (of a few hundreds of meV) are in agreement with similar systems previously investigated in

the literature.

Nevertheless, notice that in the actual fitting of the experimental curve, which is made with a superposition of Lorentzian oscillators, the fitting parameters are the oscillator strength and the linewidth of each peak, but not the exciton binding energies. The binding energies mentioned in the manuscript are just a way to understand the existence of two clear peaks in each experimentally obtained spectra. Therefore, there is no propagation of errors as a consequence of a possible imprecision of the binding energies, since these binding energies are not used in the fitting procedure.

2 The authors have provided very solid calculations to demonstrate the existence of HEPs. I highly recommend the authors add some comments on the future experiments: the possible ways to probe the dispersions of HEPs, considering the monolayer BP might be sensitive to nano-fabrications.

We sincerely thank the referee for this insightful comment. The instability of BP adds difficulties in detecting both the excitons and polaritons properties. We propose that the further experiments could be attainable via BP encapsulated within hexagonal boron nitride (hBN) multilayers made with the Van der Waals assembly technique.

For the far-field technique, one can study the polariton using periodic structure. Of course, a more attractive way is through near field imaging. The development of the real-space nanoimaging technique has allowed to directly observe these highly confined polaritons in the near field. The propagating properties and the wavelength of a polariton can be extracted from the interference fringes in the scattering-type scanning near-field optical microscopy (s-SNOM) images. The dispersion relation of the polariton is thus attainable, along with the quantitative measure of the propagation loss.

In the revised version, we add the following: “The instability of BP poses challenges in detecting the polaritons. Further experiments could be conducted on BP encapsulated by hexagonal boron nitride (hBN) through the Van der Waals assembly technique. Using scattering-type scanning near-field optical microscopy (s-SNOM), the real-space nanoimaging of hyperbolic exciton-polaritons can be performed and the polariton dispersion can be obtained by changing the incident light frequency. Alternatively, far-field absorption of periodically structured BP can also directly probe the exciton-polariton.”

3 The x-axis of Figure 3a is sort of unfriendly to readers. Replacing it with a 3D hyperbolic dispersion will be much better.

We thank the reviewer the nice suggestion. To make it clearer, we now show the ZZ and AC wave vector axes in a 3D manner in Fig. 3a.

4 In the legend of Fig3. (Row # 190). “The black solid curve represents the fitted dispersion”. The model for the fitting needs to be referred to.

The black solid curve is based on the calculated loss function, which represents the energy at which the loss function peaks for each given wave vector. To make the Fig. 3a clearer, we plot the dispersion in a 3D manner and we have added “The black dashed curve is based on the calculated loss function, which represents the energy at which the loss function peaks for each given wave vector.” in the main text.

5 All the quantitative analysis in the manuscript is from fitting the reflectance contrast spectrum. The author has to plot the fits over the raw data in Supplementary Figure 1 and list all the fitting parameters, which are very important to understand the impact of oscillator strength and linewidth.

We sincerely thank the reviewer for this suggestion. We realize the importance of the fitting process. In the previous version, we obtained the optical conductivity by assuming only three oscillators and ignore the small vibration at higher energies. In the current version, we add more oscillators to get a perfect fit and attest the reliability of the fitting results.

We have added the fitting curves over the raw data in Supplementary Figs. 1-3 of monolayer BP, few-layer BP and monolayer TMDCs. All of the spectra are modeled by a superposition of Lorentzian oscillators. An almost perfect fitting can be found in the Supplementary Figs.1-3.

To extract the oscillator strength and peak energies of the individual exciton resonances from the optical response, however, we parameterize the dielectric function with reduced number of oscillators, including the exciton ground state resonance, as well as the first excited state. This only leads to negligibly small changes in the energy range of the hyperbolic regime. The fitting curves can be found in the Supplementary Fig. 4.

The extracted fitting parameters are summarized in Supplementary Table 1 and Supplementary Table 2. The results show a relatively large oscillator strength in monolayer BP. Actually, the frequency-integrated real part of the optical conductivity (exciton absorption) is another reliable scheme to illustrate the oscillator strength. We also summarized the integrated optical conductivities among atomically thin BP and monolayer TMDCs in Supplementary Fig. 9. We added “The extracted fitting parameters, including oscillator strength, peak energy and linewidth are summarized in Supplementary Table 1 and Table 2. As a reference, Supplemental Fig. 9 plots the energy differences Δ_{12} and the integrated optical conductivities (exciton absorption) as well.” in the revised version.

Extracted parameters of 1L BP								
1s			2s			3s		
f_k (eV ²)	ω_k (eV)	γ_k (eV)	f_k (eV ²)	ω_k (eV)	γ_k (eV)	f_k (eV ²)	ω_k (eV)	γ_k (eV)
4.331	1.698	0.048	0.042	1.926	0.025	0.090	2.006	0.100
Higher transition 1			Higher transition 2			Higher transition 3		

f_k (eV ²)	ω_k (eV)	γ_k (eV)	f_k (eV ²)	ω_k (eV)	γ_k (eV)	f_k (eV ²)	ω_k (eV)	γ_k (eV)
14.802	2.010	1.100	16.282	2.745	0.912	14.562	3.074	0.382
Extracted parameters of 2L BP								
1s			2s			\		
f_k (eV ²)	ω_k (eV)	γ_k (eV)	f_k (eV ²)	ω_k (eV)	γ_k (eV)	\	\	\
1.027	1.121	0.033	0.870	1.240	0.177	\	\	\
Extracted parameters of 3L BP								
1s			2s			Higher transition		
f_k (eV ²)	ω_k (eV)	γ_k (eV)	f_k (eV ²)	ω_k (eV)	γ_k (eV)	f_k (eV ²)	ω_k (eV)	γ_k (eV)
0.630	0.846	0.022	0.136	0.945	0.072	0.473	1.038	0.25
Extracted parameters of 4L BP								
1s			2s			Higher transition		
f_k (eV ²)	ω_k (eV)	γ_k (eV)	f_k (eV ²)	ω_k (eV)	γ_k (eV)	f_k (eV ²)	ω_k (eV)	γ_k (eV)
0.212	0.694	0.017	0.050	0.776	0.067	0.204	0.857	0.220
Extracted parameters of 5L BP								
1s			2s			E ₂₂		
f_k (eV ²)	ω_k (eV)	γ_k (eV)	f_k (eV ²)	ω_k (eV)	γ_k (eV)	f_k (eV ²)	ω_k (eV)	γ_k (eV)
0.153	0.613	0.019	0.282	0.6900	0.159	0.873	1.276	0.160

Supplementary Table 1. Summary of fitted oscillator strength, peak energy, and linewidth of BP. The higher transition peak in the Supplementary Table 1 has no clear origin but just to allow the tail of spectra to be fitted. The 2s state of 2L BP is not obvious in our measurement due to the cut off energy of our setup. As a result, the fitting parameters of 2s are not so reliable.

Summary of extracted parameters						
Exciton states	1s			2s		
Parameters	f_k (eV ²)	ω_k (eV)	γ_k (eV)	f_k (eV ²)	ω_k (eV)	γ_k (eV)
1L BP	4.331	1.698	0.048	0.042	1.926	0.025
2L BP	1.027	1.121	0.033	0.870	1.240	0.177
3L BP	0.630	0.846	0.022	0.136	0.945	0.072
4L BP	0.212	0.694	0.017	0.050	0.776	0.067
5L BP	0.153	0.613	0.019	0.282	0.6900	0.168
1L MoSe ₂	2.939	1.579	0.041	\	\	\
1L WSe ₂	2.163	1.662	0.040	0.260	1.868	0.088
1L WS ₂	2.059	2.025	0.024	0.163	2.200	0.062
1L MoS ₂	1.292	1.902	0.047	0.197	2.097	0.055

Supplementary Table 2. Comparison of 1s and 2s states among BP and 1L TMDCs. The 2s state of 1L MoSe₂ is not available in our measurement.

Reviewers' Comments:

Reviewer #1:

Remarks to the Author:

First, I would like to thank the authors for trying to emphasize the main novelties of their work. As mentioned in my previous report, and as noted by the authors, the in-plane hyperbolic propagation of polaritons in nanomaterials is a hot and very interesting topic. In this regard, I find the results shown in this work promising. However, unfortunately, I still believe that the quality of the current manuscript is not at the level of Nature Communications, and I would rather recommend publication in a more specialized journal.

Once again, let me stress that while the authors suggest/predict the existence of exciton polaritons in BP for very thin layers below the limit of approx. 4-5L, whether these polaritons are useful for applications is unknown since the optical properties of such polaritons by direct means are not reported/revealed (this is especially relevant for BP in the ultra-thin limit, which is highly unstable as recognized by the authors). In addition, note that prediction of highly confined in-plane propagating exciton-polaritons has been reported on monolayer semiconductors [ref. 11 in the manuscript, 2D Mater. 7, 035031 (2020)]. As mentioned by the authors in the revised version, a direct visualization and in-depth analysis of the data would help in this regard and surely would improve the quality of the manuscript.

Reviewer #2:

Remarks to the Author:

The authors have well addressed the review/comments. I think the manuscript is in a good shape to be published in Nature communications.

Reviewer #3:

Remarks to the Author:

I find that the authors have successfully addressed all my questions and I recommend the paper for publication in Nature Communications.

Response to reviewers' comments

Reviewer #1 (Remarks to the Author):

First, I would like to thank the authors for trying to emphasize the main novelties of their work. As mentioned in my previous report, and as noted by the authors, the in-plane hyperbolic propagation of polaritons in nanomaterials is a hot and very interesting topic. In this regard, I find the results shown in this work promising. However, unfortunately, I still believe that the quality of the current manuscript is not at the level of Nature Communications, and I would rather recommend publication in a more specialized journal.

Once again, let me stress that while the authors suggest/predict the existence of exciton polaritons in BP for very thin layers below the limit of approx. 4-5L, whether these polaritons are useful for applications is unknown since the optical properties of such polaritons by direct means are not reported/revealed (this is especially relevant for BP in the ultra-thin limit, which is highly unstable as recognized by the authors). In addition, note that prediction of highly confined in-plane propagating exciton-polaritons has been reported on monolayer semiconductors [ref. 11 in the manuscript, 2D Mater. 7, 035031 (2020)]. As mentioned by the authors in the revised version, a direct visualization and in-depth analysis of the data would help in this regard and surely would improve the quality of the manuscript.

To make it clear that we did not observe the polaritons by direct means, we have modified the title to "Prediction of hyperbolic exciton-polaritons in monolayer black phosphorus". We have also modified the abstract "we demonstrate that monolayer black phosphorus naturally hosts hyperbolic exciton-polaritons" into "we predict that monolayer black phosphorus naturally hosts hyperbolic exciton-polaritons". (Page 2, line 7)

We believe our study would stimulate the exploration of exciton-polariton in monolayer BP, and further near-field imaging of hyperbolic exciton-polariton in monolayer BP would be achievable soon.

Reviewer #2 (Remarks to the Author):

The authors have well addressed the review/comments. I think the manuscript is in a good shape to be published in Nature communications.

We are extremely grateful for the very positive assessment from the reviewer.

Reviewer #3 (Remarks to the Author):

I find that the authors have successfully addressed all my questions and I recommend the paper for publication in Nature Communications.

We sincerely appreciate the very positive comment from the reviewer.